# The global population stru cture of *Lacticaseibacillus rhamnosus* and its application to an investigation of a rare case of infective endocarditis

Phillip P. Santoiemma[1], Susan E. Cohn[1], Samuel W. M. Gatesy[1], Alan R. Hauser[1,2], Saaket Agrawal[3], Maria E. Theodorou[4], Kelly E. R. Bachta[1], Egon A. Ozer[1,5]*

1 Division of Infectious Diseases, Department of Medicine, Feinberg School of Medicine, Northwestern University, Chicago, Illinois, United States of America, 2 Department of Microbiology-Immunology, Feinberg School of Medicine, Northwestern University, Chicago, Illinois, United States of America, 3 Department of Medicine, Northwestern University Feinberg School of Medicine, Chicago, Illinois, United States of America, 4 Division of Hospital Medicine, Department of Medicine, Northwestern University Feinberg School of Medicine, Chicago, Illinois, United States of America, 5 Center for Pathogen Genomics and Microbial Evolution, Havey Institute for Global Health, Northwestern University Feinberg School of Medicine, Chicago, Illinois, United States of America

* e-ozer@northwestern.edu

**Data Availability Statement:** All relevant data are within the paper and its Supporting Information files.

## Abstract

### Background

*Lacticaseibacillus* (formerly *Lactobacillus*) *rhamnosus* is widely used in probiotics or food supplements to promote microbiome health and may also be part of the normal microbiota of the human gastrointestinal tract. However, it rarely also causes invasive or severe infections in patients. It has been postulated that these infections may originate from probiotics or from endogenous commensal reservoirs. In this report, we examine the population structure of *Lacticaseibacillus rhamnosus* and investigate the utility of using bacterial genomics to identify the source of invasive *Lacticaseibacillus* infections.

### Methods

Core genome phylogenetic analysis was performed on 602 *L. rhamnosus* genome sequences from the National Center for Biotechnology public database. This information was then used along with newly generated sequences of *L. rhamnosus* isolates from yogurt to investigate a fatal case of *L. rhamnosus* endocarditis.

### Results

Phylogenetic analysis demonstrated substantial genetic overlap of *L. rhamnosus* isolates cultured from food, probiotics, infected patients, and colonized individuals. This was applied to a patient who had both consumed yogurt and developed *L. rhamnosus* endocarditis to attempt to identify the source of his infection. The sequence of the isolate from the patient's bloodstream differed at only one nucleotide position from one of the yogurt isolates. Both isolates belonged to a clade, identified here as clade YC, composed of mostly

**Funding:** The work was supported by the National Institute of Allergy and Infectious Diseases at the National Institutes of Health, URL: https://www.niaid.nih.gov/ (grant U19AI135964 to E.A.O. and A.R.H and grants R01AI118257, K24AI104831, R21AI153953, and R21AI164254 to A.R.H.) and the American Cancer Society, URL: https://www.cancer.org/ (Clinician Scientist Development Grant #134251-CSDG-20-053-01-MPC to K.E.R.B). The funders had no role in study design, data collection and analysis, decision to publish, or preparation of the manuscript.

**Competing interests:** The authors have declared that no competing interests exist.

gastrointestinal isolates from healthy individuals, some of which also differed by only a single nucleotide change from the patient's isolate.

## Conclusions

As illustrated by this case, whole genome sequencing may be insufficient to reliably determine the source of invasive infections caused by *L. rhamnosus*.

## Introduction

In recent years, probiotics have increasingly been used to promote general health and to prevent infections such as *Clostridioides difficile* colitis, bacterial vaginosis, and ventilator-associated pneumonia [1]. Particular attention has been given to species in the *Lactobacillus* group, such as *Lacticaseibacillus* (formerly *Lactobacillus*) *rhamnosus*, a common human and animal commensal species as well as one used frequently in commercial products such as foods and probiotic preparations [2, 3]. Yogurt has been promoted as a beneficial source of *Lactobacillus* probiotics [4] and several strains of *L. rhamnosus* used in yogurt and other food production are "Generally Recognized As Safe" (GRAS) by the United States Food and Drug Administration (FDA) [5–8]. Although *Lactobacillus* bacteria are usually considered non-pathogenic, more than 200 cases of severe infections have been reported, including bloodstream infections and infective endocarditis [9–12]. Of particular interest are reports suggesting that ingested *Lactobacillus*-containing foods or probiotic products can escape the gastrointestinal (GI) tract and enter the surrounding tissues or bloodstream to cause invasive infections [13, 14]. This is especially pertinent to patients with immune compromise, GI structural defects, or those who undergo diagnostic or surgical procedures that compromise the integrity of the GI tract [15–17]. Confounding this is the fact that several species of *Lactobacillus* are considered part of the normal microbiota of the GI and female genital tracts [18]. Therefore, it is possible that commensal rather than ingested bacteria may be sources of severe infection [19, 20]. A better understanding of the origins of *L. rhamnosus* causing bloodstream infections, endocarditis, or other invasive infections could help investigators understand why these develop in some patients and may help in the development of strategies to reduce the risk of such infections from occurring.

Whole genome sequencing (WGS) approaches are frequently applied to investigations of infections and outbreaks. WGS-based typing of cultured microbes allows for higher resolution and discrimination of isolates than older techniques such as pulsed field gel electrophoresis (PFGE), and random amplification of polymorphic DNA (RAPD), which are prone to both over- and underestimation of isolate relatedness and limited inter-laboratory reproducibility [21, 22]. A non-WGS sequence-based approach for assessing isolate relatedness is multi-locus sequence typing (MLST), which assesses variability across seven conserved genes. However, MLST typing schemes are often not well-described or widely available for many rare, emerging, or less well studied pathogens. In contrast, WGS provides high resolution characterization of variability across the entire genome and can readily be compared to results generated at other institutions. This portability of WGS data, combined with relevant metadata such as culture dates, geographic location, or isolation source, has the potential to define strain relatedness more precisely as well as provide the local or global context of isolates under investigation.

In this study we sought to examine the genetic population structure of *Lacticaseibacillus rhamnosus* and assess the utility of performing WGS to uncover the potential origin of the

pathogen in a rare case of invasive infection. Here, we present a phylogenetic analysis of a large number of publicly available *Lacticaseibacillus rhamnosus* isolates and characterize several major clades of closely-related genome sequences. We also assess whether an invasive human infection can be connected to food-associated *L. rhamnosus* in a clinical context by using WGS to investigate a previously reported case of fatal *Lactobacillus* endocarditis in a patient who consumed yogurt daily [23]. We demonstrate that food, probiotic, GI commensal, and infection isolates are intermixed across the phylogenetic tree and in many instances show surprisingly little inter-isolate sequence variability which could have implications on the usefulness of WGS in determining the source of *L. rhamnosus* infections in cases involving commonly encountered lineages.

## Results

### Global population structure of *L. rhamnosus*

To explore the population structure of *L. rhamnosus*, we queried the assembly and short read archive (SRA) databases of the National Center for Biotechnology Information (NCBI) to identify unique, as indicated by unique BioSample IDs, whole genome assemblies (n = 183) and Illumina read sets (n = 419) from *L. rhamnosus* whole genome sequencing projects, excluding low quality assemblies and read sets as described in Methods (S1–S3 in). Alignments of the 602 genomes against the reference genome sequence of strain ATCC 11443 were used to generate a maximum likelihood (ML) phylogenetic analysis of the *L. rhamnosus* species (Fig 1A). Clustering sequences based on phylogenetic distance identified five clades consisting of 20 or more isolates which combined accounted for 446 (74%) of all sequences. The largest of these clades included 230 closely-related isolates represented by the GG strain of *L. rhamnosus* (Fig 1A and S1A Fig), a strain that is widely used in probiotic preparations [24]. Most GG isolates were obtained from dairy (29.6%) or probiotic product (26.5%) sources (Fig 2A). Of the 58 clade GG isolates identified as originating from humans (25.2%), most were isolated from feces or other GI sources (56.9%) while 18 isolates (31%) were isolated from the blood (Fig 2B and S1 Fig). Interestingly, many clade GG isolates had nearly identical whole-genome sequences despite being cultured from different sources. A second large clade, consisting of 117 isolate sequences from the public database, was also identified and will become relevant later in the current study. For the purposes of this report, we designated this clade as "YC" (Fig 1A and 1B). Unlike the GG clade, most of the isolates in clade YC were of human origin (49.6%) with probiotic products the next most common isolation source (19.7%) (Fig 2A). Only thirteen isolates (11.1%) were identified as originating specifically from dairy products. These findings indicate that the global population structure of *L. rhamnosus* is diverse, but also contains prominent clonal groups consisting of closely-related isolates of both human and non-human origin.

### Demographic characteristics and associations among global *L. rhamnosus* isolates

We sought to investigate whether clinical or demographic features of *L. rhamnosus* isolates could be associated with the genetic population structure. Five clades with 20 or more sequences each were identified. These are highlighted in Fig 1A and include the GG (n = 230) and YC (n = 117) clades described above, as well as three smaller clades we refer to here as C1 (n = 39), C2 (n = 31) and C3 (n = 29). Among these clades there was little variability in the proportions of sequences from each of the identified isolation sources (Fig 2A), continent of isolation (Fig 2C) or year of isolation (Fig 2D) either between the major clades or compared to the

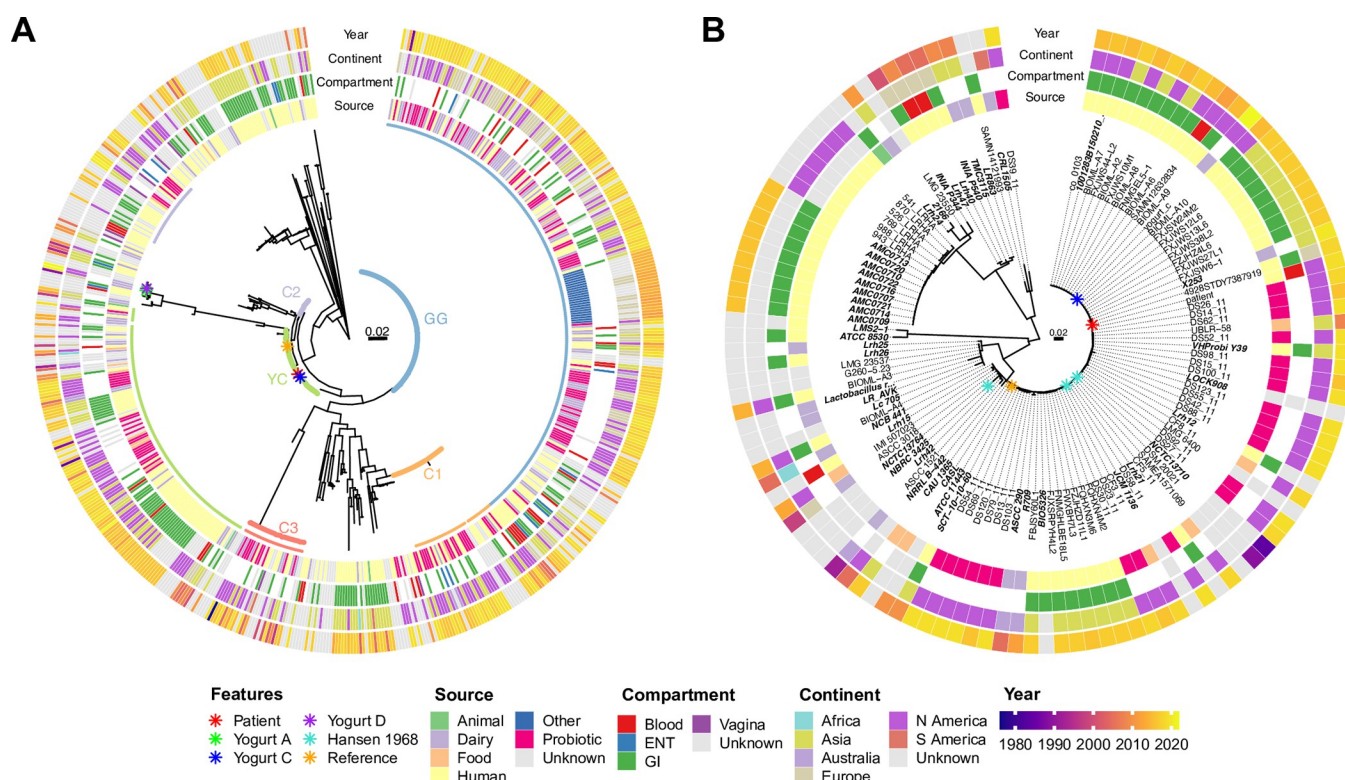

**Fig 1. Whole genome phylogenetic analysis of *Lacticaseibacillus rhamnosus*.** A) Maximum likelihood phylogenetic tree, midpoint-rooted, from whole genome alignment of *L. rhamnosus* assemblies and sequence reads from the NCBI database (n = 602) to the reference genome sequence of strain ATCC 11443 (orange star). Taxa highlighted with colored squares indicate major clades consisting of 20 or more sequences. Sequences of the case patient's bloodstream isolate ("Patient"–red star) and representative yogurt isolates ("Yogurt A, C, D"–green, blue, purple stars) are also included. B) Maximum likelihood phylogenetic tree of clade YC isolates, including case patient's bloodstream isolate and yogurt C isolates. Turquoise stars indicate "Hansen 1968" isolate sequences. In both panels, rings from inside to outside indicate, in order: 1. isolation source, 2. body site (compartment) of isolation for human sourced isolates, 3. continent of isolation, and 4. year of isolation. Taxon labels in bold indicate sequences for which only genome assemblies were available. Scale bars indicate genetic distance.

distributions among all 602 sequenced isolates ("All"). Similarly, there was little variability between clades among sequences of isolates cultured from humans (Fig 2B). Isolates from humans and probiotics were found among all clades with only clade C2 having no isolates from dairy sources (Fig 2A). Among the human isolates, the majority of sequences in all groups were obtained from gastrointestinal sources (Fig 2B). All clades had representative isolates from the continents of Asia and North America and while four out of five clades had members isolated from Europe, the largest proportion of European isolates were found within the GG clade (Fig 2C). Among all clades the majority of isolates were cultured between 2015 and 2019, but clade C3 had the largest proportion of isolates cultured prior to 2015 (Fig 2D). Overall, there was little evidence of an association of any of the clades with any particular demographic traits.

To further assess these relationships, we identified all sequences in the five clades for which source, continent, and year data were available (n = 314, 70%). Hierarchical clustering based on these three criteria largely showed a broad distribution of phylogenetic clades across the demographic clusters with few clear patterns of association (S2A Fig). One possible exception revealed by this analysis was that the large majority of sequences from dairy sources were isolated from Europe and that nearly all of these sequences were found in the GG clade. It cannot be excluded that this finding may be the result of more extensive sampling and sequencing of

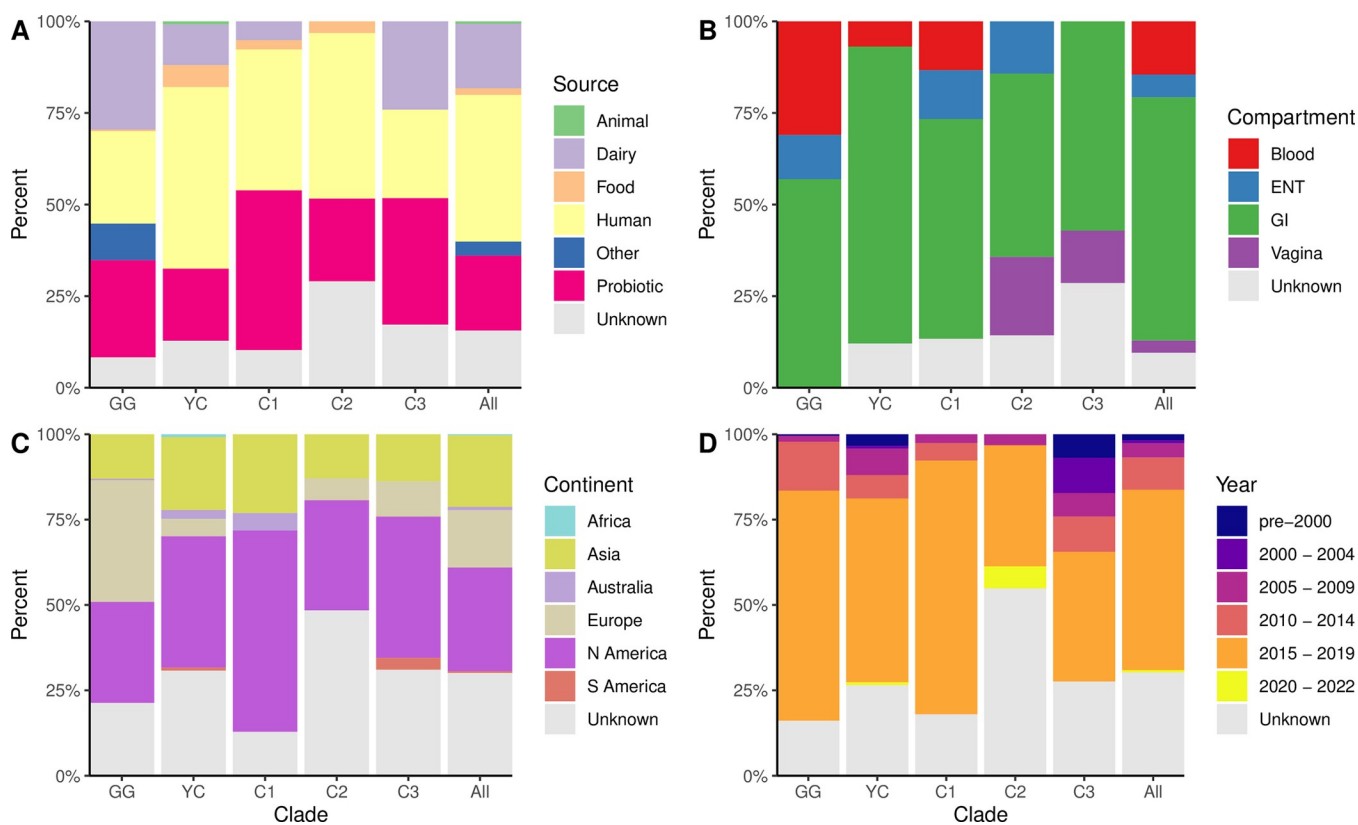

**Fig 2. Demographic characteristics of *Lacticaseibacillus rhamnosus* sequences among major phylogenetic clades.** Distributions of isolation source (A), isolation site or compartment for human-derived isolates (B), continent of isolation (C), and year of isolation (D) among isolates belonging to each of the five major clades and among the complete sequence collection ("All"). Bar heights indicate the percent of the total sequences in each group.

*L. rhamonosus* isolated from dairy sources in Europe versus a true predominance of GG clade isolates in this niche and location. Similarly, a multiple component analysis (MCA) of these data showed overlap of concentration groups consisting of sequences from all clades (S2B Fig). In summary, no clear association between genetic diversity and either source, continent, or year of isolation was identified among available *L. rhamnosus* isolate sequences.

## A case of *L. rhamnosus* bacteremia and endocarditis

To test whether the origin of a *L. rhamnosus* clinical isolate might be determined from its genome sequence, we investigated a previously reported case of *Lactobacillus* endocarditis at our institution. In December 2019, an 83-year-old male presented with fever, cough and rigors. His past medical history included transaortic valve replacement for aortic stenosis in 2017, osteoarthritis and lumbar stenosis requiring posterior spinal fusion in 2013, Crohn's Disease requiring small bowel resection in 2000, hepatitis C infection-mediated cirrhosis followed by sustained virologic response after treatment, IgG monoclonal gammopathy treated with chronic intravenous immunoglobulin (IVIG) infusions, and prostate cancer in remission. A few weeks prior to presentation, the patient underwent a dental cleaning for which he took prophylactic amoxicillin. On presentation, the patient was diagnosed with vertebral osteomyelitis and *Lactobacillus* bacteremia but had no evidence of endocarditis by transesophageal echocardiogram (TEE). The source of the bacteremia could not be definitively established in this patient. He was treated with intravenous (IV) ampicillin for 6 weeks with clinical

improvement. Two weeks after completion of this antibiotic course, he presented with a second episode of prolonged *Lactobacillus* bacteremia marked by five consecutive days of positive blood cultures. A repeat TEE indicated a new vegetation on his prosthetic aortic valve. He was treated with IV ampicillin and gentamicin and initially improved but one week later developed a large parietal intracranial hematoma presumably from rupture of a mycotic aneurysm. He subsequently decompensated, was transitioned to comfort care, and passed away in March 2020. A more detailed report of this patient's clinical case has been previously published [23].

## Multiple *Lactobacillus* species and diverse *L. rhamnosus* strains can be found among yogurt brands commercially available in the United States

The patient and his family indicated to the clinical staff that he regularly ate yogurt to promote GI health. Given that the *Lactobacillus* group of bacteria are commonly used as active culture agents in commercial yogurt products, we investigated whether there was a link between his regular yogurt consumption and his invasive *Lactobacillus* infection. To identify the *Lactobacillus* species causing the patient's infection, the genome of a bloodstream isolate from the patient's second hospitalization was sequenced. Additionally, we purchased single servings of four different yogurt brands (anonymized in this report as A, B, C, and D) from a local branch store of a North American supermarket chain. Brand A had been identified by the patient's family as one that he frequently consumed, but not exclusively. The other three brands purchased were a convenience sampling representative of yogurt brands broadly available in supermarkets in this region of the United States. Samples of each yogurt were cultured under *Lactobacillus*-selective conditions, and six cultured colonies from each brand were chosen at random (designated A-1 through 6, B-1 through 6, etc.) for whole-genome sequencing. Two of the isolates selected, A-3 and B-1, sequenced poorly and were excluded from further analyses. Analysis of the 16S rRNA gene sequences from the genome assemblies identified the patient's bloodstream isolate, as well as all isolates from brands A and D and three of the six isolates from brand C as *Lacticaseibacillus rhamnosus*. The other three isolates from brand C and all five isolates from brand B speciated as *Lacticaseibacillus paracasei*. Reference-based read alignment and phylogenetic analysis of the *L. rhamnosus* yogurt isolates revealed a clonal relationship among isolates from brands A and D (Fig 3). The three *L. rhamnosus* isolates from brand C and the patient's bloodstream isolate were also clonal. As *L. rhamnosus* isolates within each yogurt sampled were isogenic, isolates A-1, C-2, and D-1 were selected to represent each yogurt brand in further analyses. At first glance, these results suggested that a strain similar to that from yogurt C may have caused the patient's vascular infection.

## Bloodstream invasive *L. rhamnosus* is genetically similar to both a yogurt-derived isolate and a closely-related clade of globally- and temporally distributed isolates

We examined the patient's bacterial isolate and the yogurt isolates in the context of the publicly available *L. rhamnosus* genome sequences (Fig 1A). Neither the patient's bloodstream isolate nor any of the three representative *L. rhamnosus* isolates from commercial yogurt samples A, C, or D were part of the largest clade, GG. Two of the yogurt isolates, A and D, shared identical whole-genome sequences but were distinct from other sequences currently deposited in the NCBI sequence database. In contrast, the patient's isolate and the yogurt C isolate were both members of the YC clade. The patient's bloodstream isolate differed from the yogurt C isolate and several of the NCBI human GI tract isolates by only one single nucleotide variant (SNV) over more than 2.9 million aligned bases: an adenine to guanine amino acid change at position 1,753,277 relative to the sequence of the ATCC 11443 reference strain. This variation encodes

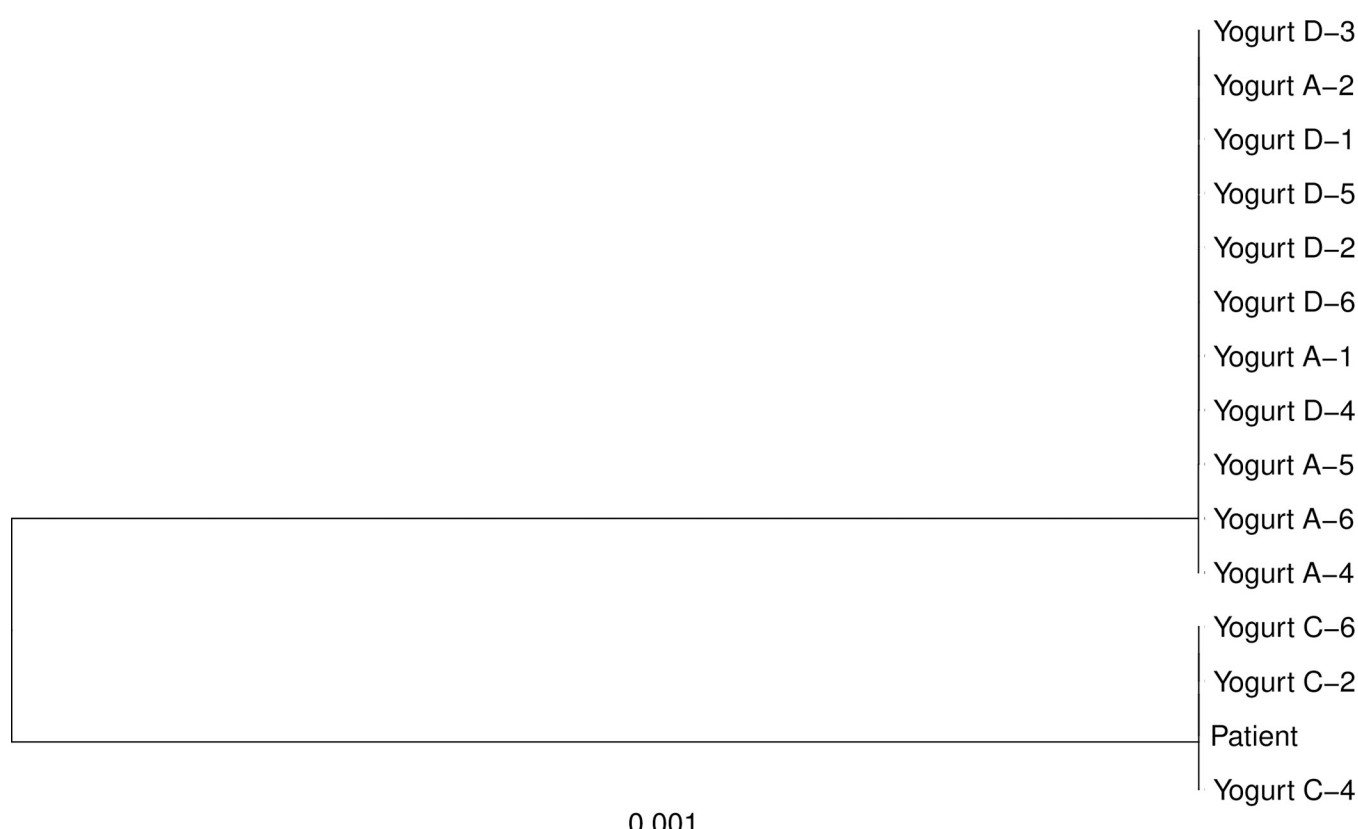

**Fig 3. Maximum likelihood whole-genome phylogenetic tree of *Lacticaseibacillus rhamnosus* sequences from a patient with bloodstream infection and sampled yogurt brands.** Sequencing reads were aligned to the genome sequence of strain ATCC 11443 and the maximum likelihood phylogenetic tree was determined from the whole genome sequence alignment. For yogurt isolates, letters (A, C, D) indicate separate yogurt brands sampled and numbers (1–6) indicate separate colonies cultured from each yogurt brand. Scale bar indicates genetic distance.

a nonsynonymous amino acid substitution, M1A, disrupting the start codon of the NrdR transcriptional regulator gene (locus ID CEA83_RS08935). None of the 602 publicly available *L. rhamnosus* genome sequences nor any of the other yogurt isolate sequences differed from the reference sequence at this position.

As the patient's isolate belonged to clade YC, a more in-depth analysis of this clade was performed. Sequence alignment and phylogenetic analysis revealed limited genomic diversity between the sequences of the patient's bloodstream isolate, the yogurt C isolate, and the 117 NCBI clade YC isolates (Fig 1B). To quantify isolate diversity within and between the largest *L. rhamnosus* clades, pairwise SNVs were counted. Average pairwise genetic distance among the 117 isolates in clade YC was 1220 SNVs (range 0–3,424), considerably higher than the average pairwise distance of 25 SNVs (range 0–261) among clade GG isolates or the average pairwise SNV distances of 18, 616, and 237 for clades C1, C2, and C3, respectively, but still much lower than the average SNV count of 44,610 (range 2,009–53,132) between sequences across the five largest clades (S3 Fig). Among the 59 isolates of human origin in clade YC, including the patient's isolate, five were identified as originating from blood, 47 were isolated from GI samples, and seven did not have the isolation site recorded (S1 Table). Of note, several of the human GI isolates as well as the previously reported bloodstream isolate SAMN12632834 were identical (0 SNV differences) to the yogurt C isolates. The oldest sequence in this clade was isolated in 1984, but the majority of sequences (53.8%) were from isolates collected between 2015 and 2019 (Fig 2D). Isolates in clade YC were globally distributed, predominantly submitted

from North America (38.5%) and Asia (21.4%) with six isolates from Australia, three from Europe, and one each from Africa and South America (Fig 2C). Of note, this clade also contained three sequences of the *L. rhamnosus* type strain "Hansen 1968" identified by different repository designations: DSM 20021, JCM 1136, and NBRC 3425 [25] (Fig 1B, turquoise stars). These three sequences were generated by different institutions and differed by between 1 and 220 total pairwise SNVs in their alignments. The two most closely related Hansen 1968 isolates, DSM 20021 and JCM 1136, differed from the yogurt C isolate by 10 and 11 SNVs, respectively. Together, these phylogenetic patterns suggest a diffuse geographical and temporal distribution of clade YC, possibly originating from the Hansen 1968 type strain, with very little or, in many cases, no genetic variation over time. Despite their genetic similarities, these isolates were cultured from multiple sources, including animals, dairy products and food, probiotics, stool from apparently asymptomatic individuals based on available sequence metadata (S1 Table), and a variety of infected sites. In summary, whole-genome sequencing could not adequately discriminate between the patient's bloodstream isolate, a yogurt isolate, and isolates presumably part of the normal microbiota of the GI tract.

## Discussion

*L. rhamnosus* is a common member of the GI normal microbiota that is also added to food and used as a probiotic. However, this normally benign bacterium is capable of causing invasive bloodstream infections and endocarditis [9–12]. Given that *L. rhamnosus* and similar species are frequent colonizers of the GI tract, it is thought that these rare infections are caused by movement of these organisms out of the gut and into adjacent blood or tissues. The mechanisms for translocation of bacteria from the gastrointestinal tract into the blood or other extraintestinal sites are thought to include intestinal bacteria overgrowth, increased gut permeability, and/or host immune deficiencies [26]. It remains controversial whether the sources of these invasive infections are from endogenous normal microbiota, i.e. commensal organisms colonizing the GI tract, or exogenous bacteria such as those found in yogurt or other dairy or probiotic products ingested before infection onset [27]. To date there have been few smaller-scale investigations of the genomic population structure of *L. rhamnosus* and associations between genetic content and niche or other characteristics, and these studies have mostly not directly examined genetic relationships between clinical- and non-clinical isolates [28–31]. Even fewer studies have sought to use WGS to associate clinical isolates of *L. rhamnosus* with potential infecting sources. One study compared genome sequences of 16 clinical blood isolates to strain LGG alone [32], and two other reports have used WGS to compare either bloodstream or invasive dental infection isolates to single strains isolated from probiotics [33, 34]. We could identify only one study that directly examined phylogenetic relationships of clinical isolates in the context of the larger *L. rhamnosus* population structure. In this study, five clinical isolates sourced from dental infections, respiratory specimens, and an abscess culture were compared to 172 publicly-available *L. rhamnosus* genome sequences and found to cluster together, but distinct from isolates collected from healthy microbiome specimens and other non-human sources [35]. To our knowledge, this is the first study to both examine the overall population structure of *L. rhamnosus* and then performed an in-depth analysis of *L. rhamnosus* isolated from the bloodstream of a patient with endocarditis to examine associations of largely food-borne (i.e. commercial yogurt) isolates with invasive and severe infection. Our results indicate that *L. rhamnosus* isolates from many different types of reservoirs and geographic sources are genetically nearly identical with our case's and others' bloodstream isolates, making definitive source identification difficult even with whole-genome sequencing.

We identified just one nucleotide difference in the patient's bloodstream isolate relative to the yogurt C isolate and other GI isolate sequences in the YC clade. This nucleotide change resulted in a nonsynonymous amino acid change disrupting the start codon of the *nrdR* transcriptional regulator gene. *ndrR* encodes a transcriptional repressor usually found clustered with ribonuclease reductase (RNR) genes or genes involved in primosome assembly and DNA replication in many bacterial species [36–38]. NdrR is thought to interact with thioredoxin and play a role in homeostasis maintenance [39]. Despite this important role, studies in other bacterial species have shown that deletion of *ndrR* did not affect growth profiles in culture, though deletion in *Pseudomonas aeruginosa* and *Streptococcus pyogenes* did impact adherence of the bacteria to host cells [40, 41]. It is unclear whether this mutation in the patient isolate provides a selective advantage in prolonged bloodstream infection or endocarditis or merely represents evolutionary drift.

It is noteworthy that *L. rhamnosus* isolates of nearly identical sequences were of human GI tract, yogurt, or other food or probiotic origin and were cultured from individuals from multiple continents. The geographic and temporal diversity of genetically similar clade YC isolates would suggest that they stem from a common source closely related to the Hansen 1968 type strain. One explanation is that commercial dairy or probiotic products manufactured around the world use this particular strain of *L. rhamnosus* and that, following ingestion, this strain is capable of at least transiently colonizing the human GI tract [13, 33, 42–49]. To our knowledge, this is the first published report defining this globally distributed clade of *L. rhamnosus* isolates.

Genomic analysis of publicly available sequences identified a second large clade of *L. rhamnosus*, the GG clade. This clade consists of isolates of *L. rhamnosus* recovered from the human GI tract, food, as well as bloodstream infections. The *L. rhamnosus* strain GG (LGG) was first isolated from fecal samples of a healthy adult and has since been widely used as a probiotic strain largely due to its acid and bile resistance, favorable growth profile, and intestinal epithelial adhesion properties [42]. LGG has also been associated with invasive disease [13, 33, 43–45]. The sequence database also revealed that, as with clade YC, GI carriage of LGG clade isolates can occur. This is supported by prior studies demonstrating recovery of *Lacticaseibacillus* GG in the GI tract or stool during or after probiotic therapy, though GI carriage was frequently not sustained for more than a few weeks after ceasing ingestion of the *Lacticaseibacillus*-containing probiotic [46–49]. Notably, study sizes in these reports were small but suggested that long-term colonization by dairy or probiotic strains of *L. rhamnosus* may depend on repeated consumption of strain-containing products.

Our analysis includes several limitations. To try to capture the breadth of diversity in the *L. rhamnosus* population structure, we sought to include as many high-quality genomic sequences as possible. Some of the whole-genome sequence assemblies we included could contain consensus errors, possibly artifactually changing the measured genetic distance and inferred relatedness of these isolates. As differing software and methodologies for assessing variants for sequencing reads and assembled genome sequences could introduce systematic bias, we sought to minimize cross-technique effects by generating pseudoreads from assemblies and employing a single shared alignment and SNV-calling method for both data types. We have highlighted the different sequence data types in Fig 1B and S1 Fig. Another limitation is that our assessment of the population structure of *L. rhamnosus* relied on an analysis of publicly available genome sequences. These isolates may have been collected and their sequences deposited under various study conditions and goals that could introduce bias into our assessment of associations between major phylogenetic clades and isolate characteristics. A recent study in a collection of 384 *L. rhamnosus* sequences reconstitutes our finding of a lack of association between phylogenetic clade and either gut or dairy isolates [31]; however future studies

involving larger unbiased collections may be necessary to draw firmer conclusions on genotype-phenotype associations in this species. No concurrent GI samples were collected from the patient with endocarditis to allow examination of possible GI carriage. We also were not able to definitively determine the specific brands of yogurt the patient preferred and so chose the yogurt brands sampled in this study based on local availability. Nevertheless, we did culture lactobacilli from four common commercial brands, and *L. rhamnosus* from one of these brands was nearly identical to the patient's bloodstream isolate. Future studies involving larger specimen collections and more clinical isolates will be needed to more fully examine and validate potential associations between food or probiotic consumption and invasive *L. rhamnosus* infections.

In summary, the population structure of *L. rhamnosus* shows that many isolates from the human GI tract are nearly identical in sequence to isolates from food and probiotics, complicating the use of WGS to definitively identify the source of strains causing invasive infections.

## Materials and methods

### Bacterial strains and growth conditions

The patient's isolate was obtained from a blood culture drawn during the patient's second hospitalization for recurrent *Lactobacillus* bacteremia. Single servings of commercial yogurt brands were purchased from a local store of a national supermarket chain. Samples were streaked onto De Man, Rogosa and Sharpe (MRS) *Lactobacillus*-selective agar (Research Products International, IL) and grown for a minimum of 48 hours at 37˚ C in anaerobic conditions.

### Whole-genome sequencing

Isolates were grown overnight in MRS broth topped with paraffin oil without shaking at 37˚C to create microaerophilic growth conditions. DNA extraction was performed using a QIAamp BiOstic Bacteremia DNA Kit (Qiagen) according to the manufacturer's instructions. Sequencing libraries were prepared with a plexWell™ 96 kit (seqWell, Beverly, MA) and sequenced on an Illumina MiSeq platform (Illumina, Inc., San Diego, CA) using a v3 reagent kit yielding 2 x 301 bp paired-end reads totaling 6.4 Gbp of sequence, with an approximate per-sample read coverage of 92x. Adapter sequences were removed and reads were quality trimmed using fastp v 0.32.2 [50] and then *de novo* assembly was performed with SPAdes v3.15.4 [51]. Assemblies were filtered to remove contigs smaller than 200 bp or with average read coverage of less than 5. Sequences were deposited in the National Center for Biotechnology Information (NCBI) database (BioProject accession number PRJNA908878).

### Public sequences

All *L. rhamnosus* genome records (taxon ID 913) available as of April 20, 2022, were downloaded from the NCBI Genome database. Of the 246 genome assembly sequences identified, 15 were excluded as low quality for assembly contig counts > 400 and/or total size exceeding two standard deviations (220,627 nt) above or below the average genome size of the entire set (2,968,492 nt). These criteria were chosen to omit the lowest quality genome sequences most likely to result in inaccurate phylogenetic analysis results. To identify publicly available *L. rhamnosus* sequencing reads, the NCBI Sequence Read Archive (SRA) database was searched for all records using the Entrez query: "Lacticaseibacillus rhamnosus"[organism] AND "WGS"[strategy] AND "Illumina"[platform]. As of May 9, 2022, this query identified 495 records in SRA. Of these records, 49 sets were excluded for consisting only of single-ended reads.

In cases where multiple separate read sets representing distinct sequencing runs were available for a single BioSample ID (n = 15), one read set was randomly selected for inclusion. Two read sets with read name errors in the downloaded files were also excluded. Eight read sets were identified by 16S rRNA gene sequence to be from species other than *L. rhamnosus* and were excluded. Finally, two read sets with genome alignment coverages of less than 80% of the reference genome length were removed to leave 419 paired read sets for analysis. The set of excluded read sets is shown in S2 Table. Average read length across the remaining 838 read sets (forward and reverse) was 143.1 bp with a minimum read length of 72 and maximum of 326. FastQC v0.11.2 was used to assess the quality of the included read sets and all 419 paired sets were found to pass filters for per-sequence quality scores. In cases where both an assembled genome and read sets were deposited under the same BioSample record (n = 39) or assembly and sequence reads for the same isolate were deposited under two separate BioSample records (n = 9), the Illumina read sets were used in the analysis (S2 Table). After the above filtering steps were conducted, the total number of sequences of unique *L. rhamnosus* isolates obtained from the NCBI GenBank and SRA databases was 602 (S1 Table). Available metadata for each assembly and read record, including host, isolation source, collection date, and geographic location of isolation, were obtained from the NCBI BioSample database.

## Single nucleotide variant identification

Pseudoreads were generated from genome assembly sequences using the pseudoreads.pl script (https://gist.github.com/egonozer/d2f5b17a0ae62e76fbe9857bb81c1dd0) that is based on functionality embedded in Snippy v4.6.0 (https://github.com/tseemann/snippy/releases/tag/v4.6.0). Sequence reads and pseudoread sets were each aligned to the reference genome sequence of ATCC 11443 (NCBI accession GCA_003433395.1) using bwa v 0.7.15 [52]. Strain ATCC 11443 was chosen as an alignment reference as it is a complete ungapped genome assembly and closely related to the patient isolate. Single nucleotide variants relative to the reference were identified using bcftools v1.9 skipping bases with base quality lower than 25, alignment quality less than 30, and using a haploid model. Variants were further filtered as previously described [53] using the bcftools_filter software (https://github.com/egonozer/bcftools_filter) to remove variants with single nucleotide variant (SNV) quality scores less than 200, read consensuses less than 75%, read depths less than five, read numbers in each direction less than one, or locations within repetitive regions (as defined by blast alignment of the reference genome sequence against itself). Consensus sequences for genomes were produced by replacing variant positions with variant bases in the ATCC 11443 sequences and replacing bases filtered using the criteria above with gap characters ("-").

## Phylogenetic analysis

For the total set of all consensus sequences, variant positions with base calls in less than 100% of isolate sequences (i.e. non-core positions) were masked with N's using ksnp_matrix_filter.pl [54]. Separate complete sequence alignments consisting only of unmasked core genome consensus sequences from isolates in the same clade as the patient's isolate (YC) or those in clade GG (S1 Table) were generated. Maximum likelihood phylogenetic trees were created from these core genome alignments with IQ-TREE v1.6.1 [55, 56]. Core genome alignments were used in the phylogenetic analysis to focus on the impacts of genomic variation at the species or clade level. Phylogenetic tree visualization and annotation was performed using R v4.2.2 and the ggtree package v3.8.0 [57, 58]. Clustering based on phylogenetic distance was performed with TreeCluster v1.0.3 using the length-based clustering method with a threshold branch length of 0.002 [59].

## Supporting information

**S1 Fig. Maximum likelihood phylogenetic tree of major clade isolates.** Maximum likelihood phylogenetic trees from whole genome alignments of sequences from clades GG (A), C1 (B), C2 (C), and C3 (D) to the reference genome sequence of strain ATCC 11443. Rings from inside to outside indicate, in order: 1. isolation source, 2. body site of isolation for human sourced isolates (compartment), 3. continent of isolation, and 4. year of isolation. Taxon labels in bold indicate sequences for which only genome assemblies were available. Scale bar indicates genetic distance.
(TIF)

**S2 Fig. Associations of sequence demographic characteristics with phylogenetic clades.** Only demographics from sequences with no missing values for source, continent, or year were used (n = 314). (A) Hierarchical clustering results using average linkage based on Gower distances among demographic characteristics Source, Continent, and Year. (B) Multiple component analysis of demographic characteristics with individuals colored by Clade membership. Ellipses indicate concentrations of individuals based on normal distribution.
(TIF)

**S3 Fig. Pairwise genetic distances between isolates in the five major *L. rhamnosus* clades.** Dots indicate all pairwise genetic distances as measured by single nucleotide variant (SNV) differences among the 230 sequences in clade GG, 177 clade YC sequences, 39 clade C1 sequences, 31 clade C2 sequences, 29 clade C3 sequences or among comparisons between all pairs of sequences across the five clades ("Diff"). Overlying box and whisker plots indicate median SNV counts (thick black line) as well as first and third quartiles. Whiskers indicate 1.5 x the interquartile range (IQR) and average SNV counts per group are indicated by black diamonds. Y-axis is plotted on the $\log_{10}$ scale.
(TIF)

**S1 Table. Sequence data and isolate characteristics.** NCBI database accession numbers and associated metadata. Assembly sequence characteristics are included for isolates with assembly data and read sequence characteristics are included for those with unassembled Illumina sequence read data.
(XLSX)

**S2 Table. Public sequence data excluded from analysis.** NCBI database accession numbers and characteristics of publicly-available sequence read data that were excluded from analysis.
(XLSX)

**S3 Table. NCBI BioSamples with both assembly and read data available.** NCBI database accession numbers for BioSample records with both assembly and sequence read data available for which only sequence read data was included in the analysis.
(XLSX)

## Acknowledgments

This work was supported by the Northwestern University NUSeq Core Facility and the Robert H. Lurie Comprehensive Cancer Center. This work was also supported in part through the computational resources and staff contributions provided by the Genomics Compute Cluster, which is jointly supported by the Feinberg School of Medicine, the Center for Genetic Medicine, and Feinberg's Department of Biochemistry and Molecular Genetics, the Office of the Provost, the Office for Research, and Northwestern Information Technology. The Genomics

Compute Cluster is part of Quest, Northwestern University's high-performance computing facility, with the purpose to advance research in genomics. We would like to thank members of the Center for Structural Genomics of Infectious Diseases (CSGID) and the Hauser laboratory for their valuable comments during numerous discussions of this work. We would also like to thank Ramon Lorenzo-Redondo for his advice and guidance on clustering analysis.

## Patient consent statement

The study is not considered Human Subjects Research under US HHS guidelines as the subject of the case described herein was deceased.

## Author Contributions

**Conceptualization:** Phillip P. Santoiemma, Susan E. Cohn, Alan R. Hauser, Kelly E. R. Bachta, Egon A. Ozer.

**Data curation:** Phillip P. Santoiemma, Susan E. Cohn, Egon A. Ozer.

**Formal analysis:** Phillip P. Santoiemma, Egon A. Ozer.

**Funding acquisition:** Alan R. Hauser, Kelly E. R. Bachta, Egon A. Ozer.

**Investigation:** Phillip P. Santoiemma, Susan E. Cohn, Samuel W. M. Gatesy, Alan R. Hauser, Saaket Agrawal, Maria E. Theodorou, Kelly E. R. Bachta, Egon A. Ozer.

**Methodology:** Susan E. Cohn, Samuel W. M. Gatesy, Alan R. Hauser, Kelly E. R. Bachta, Egon A. Ozer.

**Project administration:** Egon A. Ozer.

**Resources:** Alan R. Hauser, Kelly E. R. Bachta, Egon A. Ozer.

**Software:** Egon A. Ozer.

**Supervision:** Kelly E. R. Bachta, Egon A. Ozer.

**Validation:** Egon A. Ozer.

**Visualization:** Egon A. Ozer.

**Writing – original draft:** Phillip P. Santoiemma, Egon A. Ozer.

**Writing – review & editing:** Phillip P. Santoiemma, Susan E. Cohn, Samuel W. M. Gatesy, Alan R. Hauser, Saaket Agrawal, Maria E. Theodorou, Kelly E. R. Bachta, Egon A. Ozer.

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
