## [Decision Letter · Decision Letter 0]

5 Oct 2023

PONE-D-23-27461Global population structure of Lacticaseibacillus (formerly Lactobacillus) rhamnosusPLOS ONE

Dear Dr. Ozer,

Thank you for submitting your manuscript to PLOS ONE. After careful consideration, we feel that it has merit but does not fully meet PLOS ONE’s publication criteria as it currently stands. Therefore, we invite you to submit a revised version of the manuscript that addresses the points raised during the review process.

We look forward to receiving your revised manuscript.

Kind regards,

António Machado

Academic Editor

PLOS ONE

Additional Editor Comments:

Dear authors,

Several major concerns were posed by both reviewers, since the title, abstract, introduction, and methodologies procedures. I strongly recommend a careful revision of the manuscript wit properly answering all comments and/or suggestions of the reviewers.

Best regards,

António

Reviewers' comments:

Reviewer's Responses to Questions

**Comments to the Author**

1. Is the manuscript technically sound, and do the data support the conclusions?

Reviewer #1: Yes

Reviewer #2: No

2. Has the statistical analysis been performed appropriately and rigorously? 

Reviewer #1: Yes

Reviewer #2: No

3. Have the authors made all data underlying the findings in their manuscript fully available?

Reviewer #1: Yes

Reviewer #2: No

4. Is the manuscript presented in an intelligible fashion and written in standard English?

Reviewer #1: Yes

Reviewer #2: Yes

5. Review Comments to the Author

Reviewer #1: Comments and Suggestions for Authors

The manuscript examine the population structure of Lacticaseibacillus rhamnosus and investigate the utility of using bacterial genomics to identify the source of invasive Lacticaseibacillus infections. Although the paper is well written and structured and the experiments are well planned, some questions need to be explained or clarified, as indicated as follows:

INTRODUCTION

1. " formerly Lactobacillus" appears in the Line 4, which should be supported by references.

2. The introduction has no logic, please rearrange it.

1. Why should we study the relationship between Lacticaseibacillus rhamnosus and endocarditis, or what is the necessity, please revise and add.

2 in the introduction. Author thought WGS is not accurate enough, why should this method be used for your research?

3. At the end of the introduction, the author believes that WGS cannot identify the real source of bacteria. What is the purpose of the research after that?

RESULT

1. Please explain whether the different branches of Lactobacillus rhamnosus are related to the source.

2. The result has no logic, please rearrange it. For example, to supplement the relevant information of the genome, the relationship between different branches and isolates.

DISCUSSION

1. Please analyze the discussion section in conjunction with other references and do not continue to describe the content of the results.

2. Lacticaseibacillus rhamnosus was detected in the blood of the patient. The possible cause is not discussed, please add.

3. The relationship between Lacticaseibacillus rhamnosus and geography and origin is not clearly discussed in this paper. In addition, Lacticaseibacillus rhamnosus in blood may come from food, which is not discussed in detail in this paper. The authors may refer to other relevant studies on the genome of L. rhamnosus, such as A large-scale comparative genomics study reveals niche-driven and within-sample intra-species functional diversification in Lacticaseibacillus rhamnosus(doi.org/10.1016/j.foodres.2023.113446)

References

1. Please unify the format

Reviewer #2: TITLE: It is suggested that the authors revise the title and make it more concise and appealing.

ABSTRACT

In background, Correct to formally knowns as

Some rephrasing is needed.

What do you mean by population structure?

LAY SUMMARY

What is meant by the population structure

INTRODUCTION

In first line correct to ‘and to prevent…’

Insert reference for the statement ‘a common human and animal commensal species…’

specify which type of commercial products.

What do the authors mean by saying ‘For this reason…’?

What is meant by What do you mean by "escape from GIT" Clarify.

cite reference for the statements where ingestion of lactobacillus containing foods poses a risk for patients and sick individuals.

The use of the term strain ought to be used instead of isolates

The third paragraph of the introduction section carries statements which are more suited for results and conclusion sections, for example ‘We demonstrate that food, probiotic, GI commensal, and

infection isolates are intermixed across the phylogenetic tree and frequently show surprisingly

little strain-to-strain sequence variability. These findings suggest that WGS may not be useful in

determining the source of L. rhamnosus infections….’ Similarly, the last 4 lines of the same para are also repetitive and belong to the conclusion section and not the introduction part of the article.

Rephrase statement given against cited reference 10

RESULTS

The authors must give a clear conceptual explanation of the term's unique whole genome assemblies and Illumina read sets in the context of their rationale of doing this work and experimental design

can we conclude from the sequencing data if there is significant variability in the data (supplementary table 1) such as # of read pairs/bases, alignment coverage and genome size? or even missing for many strains... explain why this is not the case here!

Again, great variation in sequencing data provided in supplementary table 1. So, is it justifiable to interpret only 10 or 11 SNVs difference? The total number of bases were provided only for DSM. Why is it so?

The result section ends with a summary which is the main conclusion of your study? Was this not obvious already?

DISCUSSION

Explain the rationale ‘of combining WGS and assemblies…’

MATERIAL & METHODS

In the section of public sequences, the manner of excluding assembly sequences appears arbitrary, contig counts or total genome size limits are vague, must be clarified, and restated.

Clarify how ‘read sets were further filtered to randomly select one set of reads from multiples…’?

How do you compare the quality of the 419 read sets with WGS? What are the benchmarks? It would be giving this comparison in a tabulated form

no detail of illumina read sequences has been provided. What is the quality of reads? is it comparable? for example, we cannot compare 60% quality reads with 95-100%

In the section Single nucleotide variant identification, ‘What is meant by SNV quality of scores of less than 200, selection of read consensuses, read depths?

In the section Phylogenetic analysis, what are separate complete sequence alignments. What is the basis of this consensus building?

In the section Patient Consent Statement, the authors make some conflicting statements regarding patient consent, they state that patient consent is not possible because of death of the patient but in the same paragraph, they also claim that it is not a requirement.

Kindly provide reference or describe the exact rule which supports that the ‘the study is not considered human subject's research

6. PLOS authors have the option to publish the peer review history of their article (what does this mean?). If published, this will include your full peer review and any attached files.

Reviewer #1: No

Reviewer #2: No

---

## [Author Response · Author response to Decision Letter 0]

3 Jan 2024

We thank the editor and reviewers for their insightful and constructive comments and suggestions. Please see the attached Response to Reviewers document for detailed point-by-point responses.

---

## [Decision Letter · Decision Letter 1]

30 Jan 2024

PONE-D-23-27461R1The global population structure of Lacticaseibacillus rhamnosus and its application to an investigation of a rare case of infective endocarditis.PLOS ONE

Dear Dr. Ozer,

Thank you for submitting your manuscript to PLOS ONE. After careful consideration, we feel that it has merit but does not fully meet PLOS ONE’s publication criteria as it currently stands. Therefore, we invite you to submit a revised version of the manuscript that addresses the points raised during the review process.

We look forward to receiving your revised manuscript.

Kind regards,

António Machado

Academic Editor

PLOS ONE

Journal Requirements:

Additional Editor Comments:

Dear authors,

I am pleased to inform you that one reviewer already endorsed the revised manuscript and the second reviewer (Reviewer 3) requested major revisions for future publication endorsement. Please carefully answer the reviewer 3's concerns and rectify the manuscript following the reviewer's comments.

Thank you and best regards,

António Machado

Reviewers' comments:

Reviewer's Responses to Questions

**Comments to the Author**

1. If the authors have adequately addressed your comments raised in a previous round of review and you feel that this manuscript is now acceptable for publication, you may indicate that here to bypass the “Comments to the Author” section, enter your conflict of interest statement in the “Confidential to Editor” section, and submit your "Accept" recommendation.

Reviewer #1: (No Response)

Reviewer #3: (No Response)

2. Is the manuscript technically sound, and do the data support the conclusions?

Reviewer #1: (No Response)

Reviewer #3: Partly

3. Has the statistical analysis been performed appropriately and rigorously? 

Reviewer #1: (No Response)

Reviewer #3: N/A

4. Have the authors made all data underlying the findings in their manuscript fully available?

Reviewer #1: (No Response)

Reviewer #3: Yes

5. Is the manuscript presented in an intelligible fashion and written in standard English?

Reviewer #1: (No Response)

Reviewer #3: No

6. Review Comments to the Author

Reviewer #1: (No Response)

Reviewer #3: The article written is important for the area of probiotics and the conclusions obtained may generate controversy, mainly because the text establishes a direct relationship between the presence of the Lactobacillus species in the patient's bacteremia and in yogurts that are of mass public consumption. Therefore, it is important to carry out an exhaustive review of the text and of the statements that are being used. In addition to this, it is important to make the following changes:

- Replace the terminology "flora" (in disuse) by microbiota.

- Include more bibliographic references in lines 56, 86-90, 92, 274-275 (original manuscript) and in all the sentences that are convenient to better support the statements that are made

- Restructure the results to make them more understandable, including subtitles to separate the various analyses performed.

- Indicate whether the yogurt brands analyzed include the one that the patient consumed (based on the relatives' statement) and justify why these brands were chosen (were they randomly selected? How can one justify the selection of 4 yogurt brands as a significant sample of all the yogurt brands offered on the market?)

- In the abstract the "core genome" is mentioned, but nothing else is indicated in the rest of the paper. Justify why the analysis was done only on the basis of the core genome, how it was defined and what the core genome comprises.

- In the discussion, cite other studies that have performed similar analyses or indicate that this is the first one.

- And finally, clarify that the condition of the patient from whom the strain was obtained was very critical and his decompensation is due to several factors. Also clarify that more analyses like this one (and even with greater statistical support) are needed to affirm that there is a relationship between probiotic consumption and the generation of serious infections.

7. PLOS authors have the option to publish the peer review history of their article (what does this mean?). If published, this will include your full peer review and any attached files.

Reviewer #1: No

Reviewer #3: **Yes: **Sandra Pamela Cangui Panchi

---

## [Author Response · Author response to Decision Letter 1]

16 Feb 2024

Reviewer #3: The article written is important for the area of probiotics and the conclusions obtained may generate controversy, mainly because the text establishes a direct relationship between the presence of the Lactobacillus species in the patient's bacteremia and in yogurts that are of mass public consumption. Therefore, it is important to carry out an exhaustive review of the text and of the statements that are being used. In addition to this, it is important to make the following changes:

1. Replace the terminology "flora" (in disuse) by microbiota.

We thank the reviewer for highlighting the update in terminology. All instances of “flora” in the manuscript have been replaced with “microbiota”.

2. Include more bibliographic references in lines 56, 86-90, 92, 274-275 (original manuscript) and in all the sentences that are convenient to better support the statements that are made

We have added more supporting references throughout the manuscript as requested. These include:

Line 84, references 5-8

Line 89, references 13-14

Line 91, references 15-17

Line 94, reference 19-20

Line 293, reference 27 

3. Restructure the results to make them more understandable, including subtitles to separate the various analyses performed.

Subtitles have been added to the Results section to better separate the study findings and some sections have been rearranged to try to improve readability and flow.

4. Indicate whether the yogurt brands analyzed include the one that the patient consumed (based on the relatives' statement) and justify why these brands were chosen (were they randomly selected? How can one justify the selection of 4 yogurt brands as a significant sample of all the yogurt brands offered on the market?)

During the course of his illness the patient had shared with clinicians that he would eat yogurt frequently to improve his gut health but was not asked at the time which brand or brands he was consuming. During the critical phase of his illness when he was no longer able to answer questions, his family was asked about his yogurt preferences. They were not able to definitively recall his preferences, but named one brand as a yogurt brand that the patient would frequently consume. They could not state with confidence that this brand was exclusively consumed by the patient, nor whether he consumed it more frequently than other brands. This brand was included in our study and is anonymized as “Brand A” in the report. The other three brands purchased were a convenience sample selected from among other brands available in the supermarket at the same time that “Brand A” was purchased. These four brands are representative of nationally-available yogurt brands that are also broadly available from supermarkets in our region of the U.S. We have updated the manuscript to better explain the rationale for the yogurt brand studied (line 214-217). 

5. In the abstract the "core genome" is mentioned, but nothing else is indicated in the rest of the paper. Justify why the analysis was done only on the basis of the core genome, how it was defined and what the core genome comprises.

We thank the reviewer for pointing out the lack of clarity on this portion of the analysis. We have updated the “Phylogenetic analysis” section in the Methods to more clearly highlight how the core genome alignments were produced for phylogenetic analysis (Line 446). 

6. In the discussion, cite other studies that have performed similar analyses or indicate that this is the first one.

We have revised and updated the first paragraph of the Discussion section to highlight the novelty of our study. Briefly, there have been few published reports of the population structure of L. rhamnosus and only one (Nadkarni et al, BMC Genomics 2020) that directly examined the genomic context of clinical L. rhamnosus isolates relative to the larger population structure. This prior study did not include any isolates associated with invasive bloodstream infection and found that the five clinical isolates studied were phylogenetically distinct from non-human source isolates. Our report is the first study to examine the phylogenetic context of a bloodstream invasive isolate and directly show that it is related to an isolate from a commercial yogurt source. It is also the first study we are aware of that describes the YC clade of globally distributed human- and non-human-associated isolates closely related to the Hansen 1968 type strain. 

7. And finally, clarify that the condition of the patient from whom the strain was obtained was very critical and his decompensation is due to several factors. Also clarify that more analyses like this one (and even with greater statistical support) are needed to affirm that there is a relationship between probiotic consumption and the generation of serious infections. 

We thank the reviewer for these suggestions. 

With regards to the first comment, while the patient’s underlying conditions, including his age, history of prosthetic heart valve replacement, IgG monoclonal gammopathy, and his history of liver cirrhosis may have collectively predisposed him to a more severe outcome from infection, the patient’s decompensation and death was very likely directly attributable to the L. rhamnosus infection as indicated by his recurrent and prolonged bacteremia, the echocardiogram evidence of aortic valve vegetation, and his ultimate intracranial hematoma most likely caused by migration of infectious material from his infected heart valve to his brain. Given that a more detailed report of the patient’s case was previously published (Agrawal S et al, BMJ Case Rep. 2020), we present an abbreviated summary of the case in this manuscript and refer to the comprehensive previous report for more details on the patient’s history and condition. 

To the second comment, we have added this as a potential limitation to the Discussion section (line 367).

---

## [Decision Letter · Decision Letter 2]

6 Mar 2024

The global population structure of Lacticaseibacillus rhamnosus and its application to an investigation of a rare case of infective endocarditis.

PONE-D-23-27461R2

Dear authors,

I am pleased to inform you that the revised manuscript was accepted for publication by both reviewers.

Thank you for submitting your work to the PLOS ONE journal and best regards,

António Machado

Reviewers' comments:

Reviewer's Responses to Questions

**Comments to the Author**

1. If the authors have adequately addressed your comments raised in a previous round of review and you feel that this manuscript is now acceptable for publication, you may indicate that here to bypass the “Comments to the Author” section, enter your conflict of interest statement in the “Confidential to Editor” section, and submit your "Accept" recommendation.

Reviewer #3: All comments have been addressed

2. Is the manuscript technically sound, and do the data support the conclusions?

Reviewer #3: Yes

3. Has the statistical analysis been performed appropriately and rigorously? 

Reviewer #3: Yes

4. Have the authors made all data underlying the findings in their manuscript fully available?

Reviewer #3: Yes

5. Is the manuscript presented in an intelligible fashion and written in standard English?

Reviewer #3: Yes

6. Review Comments to the Author

Reviewer #3: (No Response)

7. PLOS authors have the option to publish the peer review history of their article (what does this mean?). If published, this will include your full peer review and any attached files.

Reviewer #3: No
